# Development and organization of the retinal orientation selectivity map

Dominic J. Vita[1], Fernanda S. Orsi[1], Nathan G. Stanko[1], Natalie A. Clark[1] & Alexandre Tiriac [1,2,3] ✉

Orientation or axial selectivity, the property of neurons in the visual system to respond preferentially to certain angles of visual stimuli, plays a pivotal role in our understanding of visual perception and information processing. This computation is performed as early as the retina, and although much work has established the cellular mechanisms of retinal orientation selectivity, how this computation is organized across the retina is unknown. Using a large dataset collected across the mouse retina, we demonstrate functional organization rules of retinal orientation selectivity. First, we identify three major functional classes of retinal cells that are orientation selective and match previous descriptions. Second, we show that one orientation is predominantly represented in the retina and that this predominant orientation changes as a function of retinal location. Third, we demonstrate that neural activity plays little role on the organization of retinal orientation selectivity. Lastly, we use in silico modeling followed by validation experiments to demonstrate that the over-represented orientation aligns along concentric axes. These results demonstrate that, similar to direction selectivity, orientation selectivity is organized in a functional map as early as the retina.

The mammalian retina is a sophisticated sensory tissue that detects various features of the visual world before transmitting this information to the rest of the visual brain[1,2]. Among its many functions, the ability to selectively respond to the orientations and directions of moving objects—properties known as orientation (or sometimes axial) selectivity and direction selectivity, respectively—plays a crucial role in visual perception and behavior. Previous studies focusing on direction selectivity challenged the notion that this computation is homogeneously organized in the retina and is instead organized in a map of optic flow axes in the mouse retina[3,4]. In fact, it is becoming increasingly evident that the mouse retina exhibits many region-specific maps for various anatomical and functional features[2,5,6], and that location-dependent feature decoders may reflect a fundamental organizing principle that most efficiently detects important aspects of the visual scene[7]. Even though orientation selectivity is such a fundamentally well studied visual computation, how this computation is organized in the retina is unknown.

Orientation selectivity is defined by a cell's characteristic to respond strongest to a particular stimulus orientation (e.g., vertical) while responding weakest to the orientation orthogonal to the preferred (e.g., horizontal)[8,9]. Both amacrine cells[10–12] and retinal ganglion cells[10,13–15] can compute orientation selectivity, and several studies have elucidated the various subtypes, cellular mechanisms, and synaptic mechanisms of retinal orientation selectivity[10,14–16]. Traditionally, orientation selectivity in the retina is thought to be organized along location-independent cardinal axes, but this has not yet been tested with a high-throughput dataset and does not align well with current understandings of orientation selectivity maps in retinal boutons and orientation selectivity maps in the superior colliculus[17]. Downstream of the retina, orientation selectivity in the superior colliculus is organized in a location-specific manner, including both neurons of the superior colliculus and the axon terminals of retinal ganglion cells[17–19]. In mouse primary visual cortex, orientation selectivity exhibits a more salt-and-pepper organization, distinct from the mapping found in cats and

[1]Department of Biological Sciences, Vanderbilt University, Nashville, TN 37232, USA. [2]Vanderbilt Brain Institute, Vanderbilt University, Nashville, TN 37232, USA. [3]Department of Ophthalmology and Visual Sciences, Vanderbilt University, Nashville, TN 37232, USA. ✉e-mail: alexandre.tiriac@vanderbilt.edu

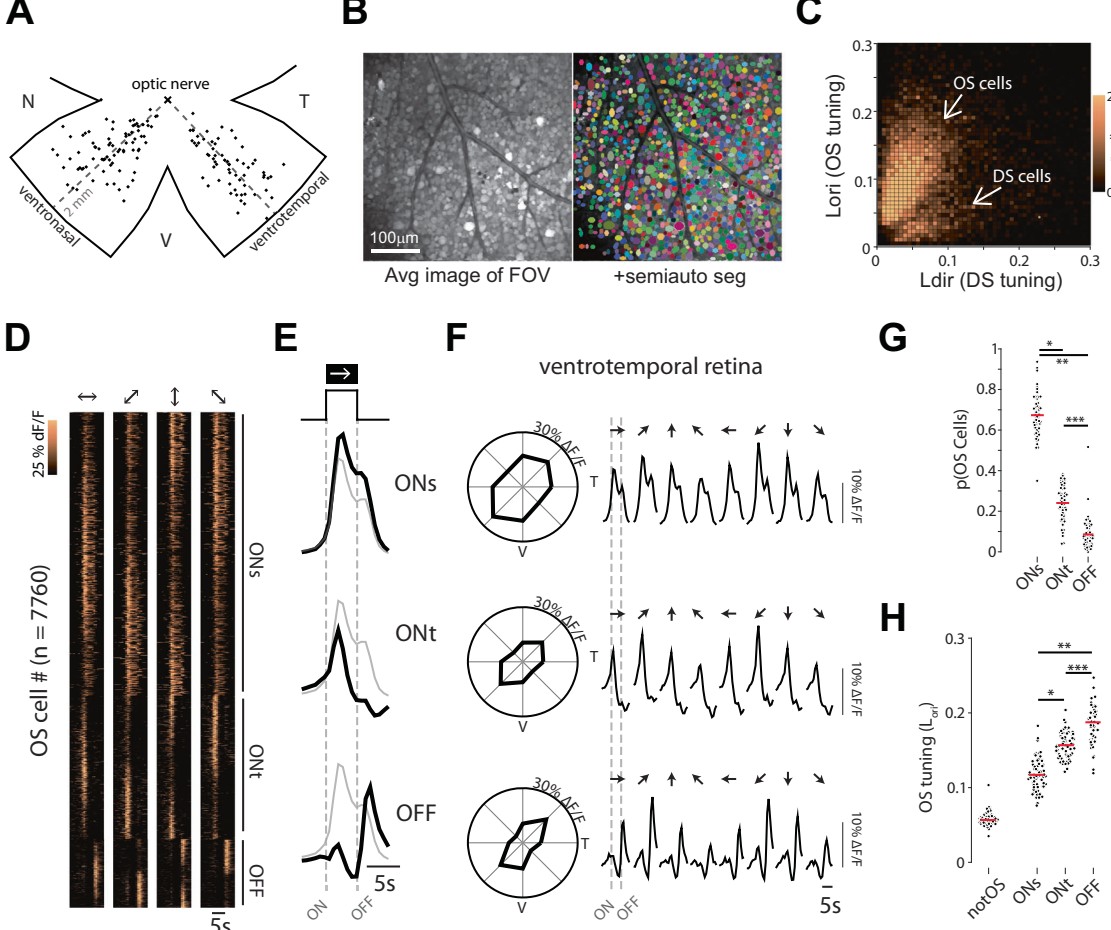

**Fig. 1 | The mouse retina has three major functional groups of orientation selective neurons. A** Schematic of ventronasal and ventrotemporal retina demonstrating the location of each imaging field of view (FOV). Each FOV (data points on panel) is $425 \times 425\,\mu m^2$ and consists of $590 \pm 170$ segmented neurons (avg ± stdev). T temporal, N nasal, V ventral. **B** Average projection of an example FOV and the results of a semiautomatic pipeline to segregate individual neurons. **C** Heatmap showing orientation selectivity tuning ($L_{ori}$) vs. direction selectivity tuning ($L_{dir}$) of all statistically significantly orientation-selective (OS) and direction-selective (DS) neurons in all FOVs. See "Methods" for detailed description of tuning calculations. **D** Heatmap showing the responses of all OS cells in both ventronasal and ventrotemporal retina, sorted based on their preferred orientation and functional subtypes (ONs: ONsustained, ONt: ONtransient, and OFF). **E** Average response to light onset and offset for each of the 3 subtypes (ONs, ONt, and OFF).

The gray trace is the average response of the entire OS population. **F** Left: tuning curves of average peak response for each of the 3 functional OS subtypes in ventrotemporal retina. Right: average response for each of the 3 functional OS subtypes. Top traces: ONs, middle trace: ONt, bottom trace: OFF. **G** Proportion of OS subtypes among all OS cells only. Each data point represents the proportion of a cell's subtype within one FOV ($N = 67$ FOVs). $*p = 1.3 \times 10^{-73}$, $**p = 4.2 \times 10^{-96}$, $***p = 4.2 \times 10^{-20}$. **H** Same as (**E**) but comparing tuning strength ($L_{ori}$) of all OS subtypes and the non-OS cells. Each data point represents the average of a subtype's tuning strength within one FOV ($N = 67$ FOVs). The "notOS" distribution depicts the tuning strength of all cells that were not statistically significantly OS. $*p = 1.6 \times 10^{-17}$, $**p = 1.3 \times 10^{-34}$, $***p = 5.5 \times 10^{-10}$, one way ANOVA followed by two-tailed Tukey test. Source data are provided as a Source Data file.

primates[20–23]. However, during development, cortical orientation selectivity exhibits signs of map organization[24].

Additionally, the organization of orientation selectivity in higher order brain regions exhibits examples of activity-dependence. In the superior colliculus, the orientation selectivity maps align to the propagation flow of retinal waves[25], which are spontaneous waves of activity generated in the retina that spread through the visual system during development. Given that retinal waves impact the development of retinal[4] and collicular[26] direction selectivity (the latter known to depend on the propagation direction of waves) as well as collicular orientation selectivity[27], it is possible that the organization of retinal orientation selectivity also depends on retinal waves. In primary visual cortex, circuits involved in orientation selectivity exhibit map refinement during development[24,28] and there is some evidence that spontaneous activity drives this refinement[24]. Whether the organization of retinal orientation selectivity depends on visual experience or spontaneous activity during development is unknown.

Here, we explore the development and organization of the retinal orientation selectivity map by using an open-access dataset consisting of large areas of the mouse ventral retina that had been presented with moving bars of light to investigate the direction selectivity map (Fig. 1A)[4]. Critically, the same visual stimulation paradigm has been used to successfully assess the various retinal orientation selectivity subtypes in adult mice[10], making this dataset ideal for a large-scale study of the orientation selectivity map in the adult and developing retina. Our investigation identifies known functional groups of orientation selective neurons and we characterize their general organization principles in the mouse retina. Importantly, we reveal a map of retinal orientation selectivity, where the preferred orientation changes as a function of retinal location. We show that this organization develops independently of retinal waves and visual experience. Lastly, we provide an initial model based on our analysis showing that retinal orientation selectivity is organized along concentric axes.

## Results

To ensure we captured the largest number of cells and thus the most accurate interpretation of functional orientation selectivity in the retina, we developed a semi-automated analysis pipeline to identify all the neurons in the ganglion cell layer of each field of view (FOV; Fig. 1B). This pipeline employs the trainable, generalist segmentation algorithm Cellpose 2.0, which utilizes an interactive machine learning approach to automatically and accurately segment hundreds of cells[29]. With this strategy, we generated a dataset of ~140,000 neurons (which includes both retinal ganglion cells and amacrine cells) across all experimental conditions allowing for an in-depth investigation of the development and organizational principles of retinal orientation selectivity.

Following segmentation of neurons, we calculated the direction and orientation/axial selectivity of each neuron using two established protocols, vector space and index calculations (refs. 10,30, see "Methods" for details). Using these calculations, we were able to identify both direction and orientation selective neurons. Moreover, our analysis successfully distinguished orientation selective cells from the direction selective ones (Figs. 1C and S1) and revealed that approximately 30% of cells in the ganglion cell layer are orientation selective (compared to 12% of cells that are direction selective; <1% of neurons were both orientation and direction selective). Importantly, we found similar results regardless of calculation methods (Fig. S1). We also ensured that our regions-of-interest (ROI) segmentation was not leading to cross contamination between ROIs (Fig. S2).

### Functional organization of orientation selectivity in the mouse retina

Orientation selective retinal cell subtypes can be distinguished by their light onset/offset preferences[10,14,15]. Performing a silhouette analysis on all orientation selective neurons revealed three functional groups based on the feature of response to light onset/offset (Figs. 1D–F and S3A, B). One class of orientation selective neuron responded transiently to light decrements with calcium signals characterized by a swift rise and decay following light offset. These OFF transient (OFF) cells made up ~10% of the total orientation selective cells and were the least represented among all three types. Another class of orientation selective neuron responded transiently to light increments with calcium signals characterized by a swift rise and decay following light onset. These ON transient (ONt) cells made up ~20% of the total orientation selective cells. The last class of orientation selective neuron exhibited a sustained response to the presence of light with calcium transients characterized by a swift rise following light onset and a pronounced decay following light offset. These ON sustained (ONs) cells made up ~70% of the total orientation selective cells and were the most represented among all three types. Interestingly, the tuning properties of these cell types were inversely related to their prevalence: OFF neurons were the most tuned, followed by ONt, and lastly ONs (Fig. 1G, H). We found this general organization of proportion and tuning to be true if we isolated our analysis to either ventronasal and ventrotemporal retina (Fig. S4). Our identified functional subtypes are consistent with previously reported OS subtypes in the mouse retina[10].

We identified that the preferred orientations of orientation selective retinal cells are organized as a function of retinal location (Figs. 2 and S5). For example, in ventronasal retina, there is an overrepresentation of cells that respond strongly to the ventrotemporal to dorsonasal axis (Fig. 2 left column). By contrast, in ventrotemporal axis, there is an overrepresentation of cells that respond strongly to the ventronasal to dorsotemporal axis (Fig. 2 right column). This mapping of preferred orientation was true for the whole population of orientation selective cells as well as for each of the functional subtypes (ONs, ONt, and OFF) that we identified. This mapping was also true if we altered the criteria used to identify orientation selective cells (Fig. S6). Although subtle, silhouette analysis revealed a second

functional group orthogonal to the overrepresented orientation (Figs. S3C, D and S6). No differences in tuning were identified between subtypes that respond to different orientations (Fig. S4).

Since this dataset used a calcium dye introduced to all cells in the ganglion cell layer, it included both retinal ganglion cells and amacrine cells, both of which are known to have subtypes that are orientation selective. With this in mind, we tested to what extent the identified orientation selective neurons represent retinal ganglion cells. To do so, we analyzed a separate dataset consisting of 26,372 neurons across 50 ventral and temporal fields of view from 4 vlgut2cre::GCaMP6s mice, which, in the ganglion cell layer, only express GCaMP6s in retinal ganglion cells. Using the same semi-automated pipeline, we performed similar analyses as above and obtained 4413 statistically significant orientation selective neurons. Consistent with the calcium dye dataset, we found similar numbers of orientation selective cells per area of the retina (Fig. S7A; $115 \pm 47$ cells per FOV in the calcium dye dataset vs. $99 \pm 42$ cells per FOV in the GCaMP6s dataset, mean $\pm$ stdev). Furthermore, we identified the same three functional groups to light onset/offset (ONs, ONt, and OFF; Fig. S7B) and similar ventrotemporal mapping as in the calcium dye dataset ($n = 298$ OS cells across 4 FOVs, 1 mouse, Fig. S7C). Lastly, within any given area of the retina, we observed an overrepresented orientation matched across all functional subtypes and another underrepresented orientation orthogonal to the overrepresented orientation (Fig. S7D). Collectively, these results suggest that orientation selective retinal ganglion cells are organized in the retina following defined parameters based in part on their preferred orientation and tuning strength.

### Retinal waves and visual experiences do not contribute to the organization of retinal orientation selectivity

The retina is spontaneously active during development and this activity[31], termed retinal waves, is involved in the development of retinal and collicular direction selectivity[4,26,27]. Importantly, orientation selectivity and organization in the superior colliculus are perturbed[27] in the β2-nAChR-KO mouse, a mouse model of reduced embryonic[32] and cholinergic[33,34] waves. Thus, with a general understanding of how orientation selectivity is organized and tuned in the adult mouse retina, we explored the role of activity on the development of retinal orientation selectivity. Retinal waves occur during three distinct stages (1, 2, and 3) between embryonic development and eye opening at postnatal day 14. Stage 1 and 2 waves are governed predominantly by cholinergic activity[32], while stage 3 waves are propagated via glutamate signaling[35] and can also be influenced by light passing through the eyelid[36]. We used data collected from mice that exhibit drastically reduced embryonic[32] and cholinergic[33,34] retinal waves (β2-nAChR-KO mice). We also used data collected from mice that were dark-reared (DR) from birth, which is a model of visual deprivation and reduced glutamatergic waves[36]. Combined, these mouse models test the roles of all stages of retinal waves and visual experience on the development of retinal orientation selectivity.

All experimental groups (normal-reared (NR), β2-nAChR-KO, and DR) had near identical organization of retinal orientation selectivity (Fig. 3). All three experimental groups exhibited an overrepresented orientation along the ventrotemporal axis in ventronasal retina, along the ventronasal axis in ventrotemporal retina (Fig. 3A), and a second underrepresented orientation. Even within a retinal quadrant, the preferred orientation of orientation selective cells changed as a function of retinal location, deviating farther away from the cardinal ventral axis (straight down) the farther cells were from the optic nerve (Fig. 3B). Using the response to light onset and offset, we identified the same 3 functional subtypes (ONs, ONt, and OFF) across all experimental conditions (Fig. 3C). We used this functional segregation to demonstrate that all experimental groups exhibited the same inverse relationship between the proportion of orientation selective subtypes and their tuning (e.g., in all groups, OFF orientation selective cells were

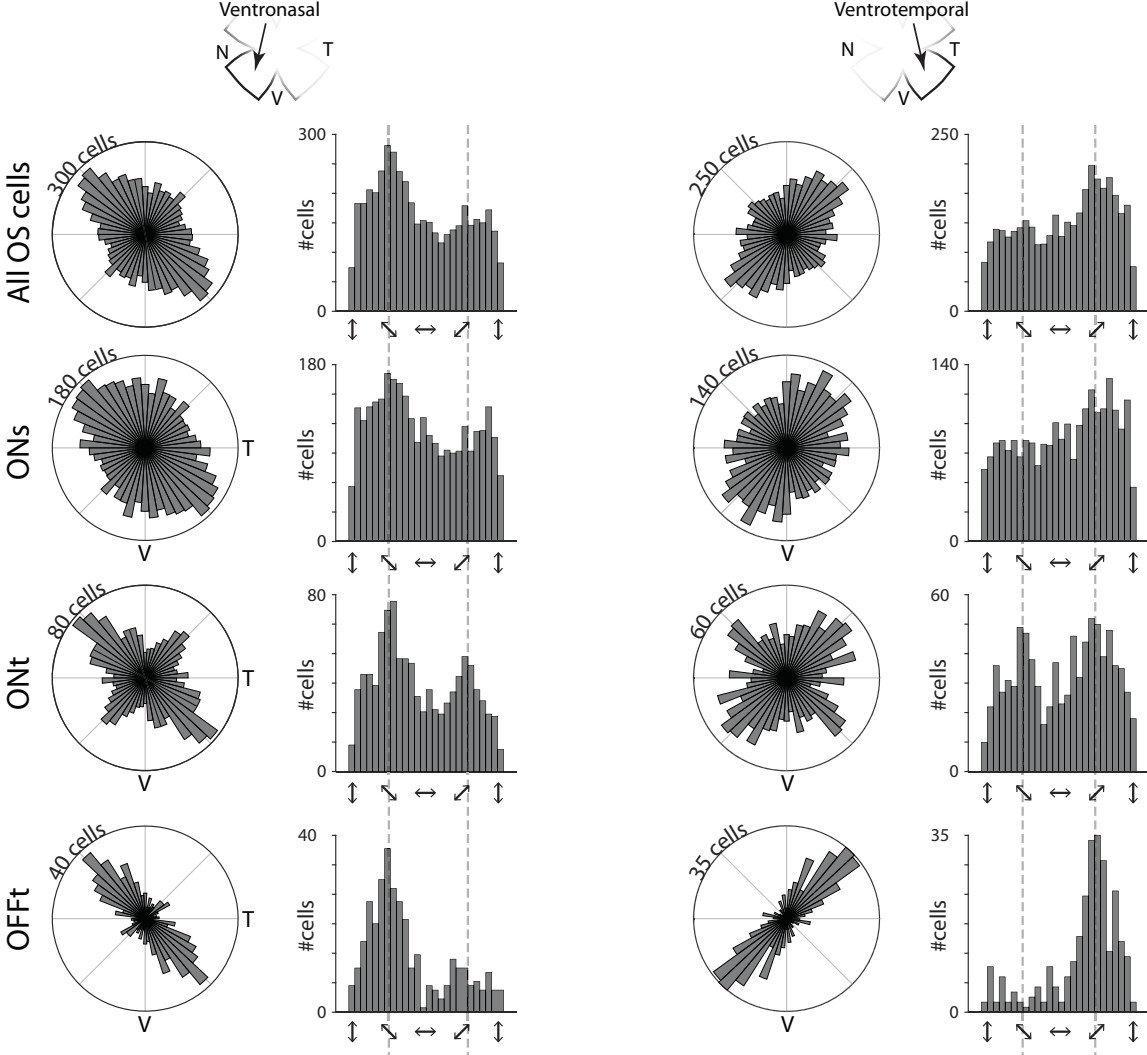

**Fig. 2 | Organization of orientation selectivity changes as a function of retinal location.** Polar and linear histograms depicting the distributions of preferred orientations in ventronasal (left column) and ventrotemporal (right column) retina for all orientation-selective (OS) cells (top row), ON sustained cells (2nd row), ON transient cells (3rd row), and OFF transient cells (4th row).

the least common and yet the most tuned; Fig. 3D). Critically, we did not find significant main effects or interactions of experimental condition (NR, β2-KO, and DR) for either proportion or tuning of OS cells. Thus, we conclude that spontaneous activity and visual experience play little to no role on the development of retinal orientation selectivity.

### Retinal orientation selectivity is organized along concentric axes

We next aimed to use our dataset to create a model of the orientation selectivity map, defined as how the preferred orientation axis varies across the surface of the retina, analogous to what has been done for the direction selectivity map[3,4]. Given how reproducible our results were across experimental groups (NR, β2-KO, and DR; Fig. 3), we pooled data across groups in order to obtain the most accurate representation of the retinal orientation selectivity map. In both ventronasal and ventrotemporal retina, the overrepresented orientation aligns along diagonal axes to the cardinal directions, or about 45° offset from ventral (Figs. 4A and 2A). This preferred orientation was reminiscent of the vertical direction selectivity preferences in ventronasal and ventrotemporal retina (see Fig. 4A and ref. 4). However, a close analysis revealed that the overrepresented preferred

orientations deviate farther from the cardinal ventral axis than ventral-preferring direction selective cells (Fig. 4B), providing initial evidence that the orientation and direction selectivity maps are distinct.

To better understand the organization and mapping of orientation selectivity, we used an in silico model to determine which axes best predict the measured preferred orientations (Fig. 4C). Previous studies report that orientation selectivity downstream of the retina is organized along concentric axes[17-19], prompting us to begin with concentric circles anchored on the optic nerve. Such an organization predicts a static ±45° preferred orientations (− in ventronasal retina, + in ventrotemporal retina) regardless of distance from the optic nerve (Fig. 4C top left). Although these parameters accounted for the average ±45° deviation from the cardinal ventral axis, they failed to account for the fact that the preferred orientation changed as a function of distance from the optic nerve. Anchoring the concentric circles in dorsal retina predicted preferred orientations whose angles greatly differed from our dataset (Fig. 4C bottom left) whereas anchoring the concentric circles in ventral retina predicted angles that much more closely aligned with the measured preferred orientations (Fig. 4C top right). Shifting the anchor slightly to ventronasal retina and using ellipses (horizontal radius 1.09x that of vertical radius) instead of circles improved the fit (Fig. 4C, bottom right). Hence, we conclude that

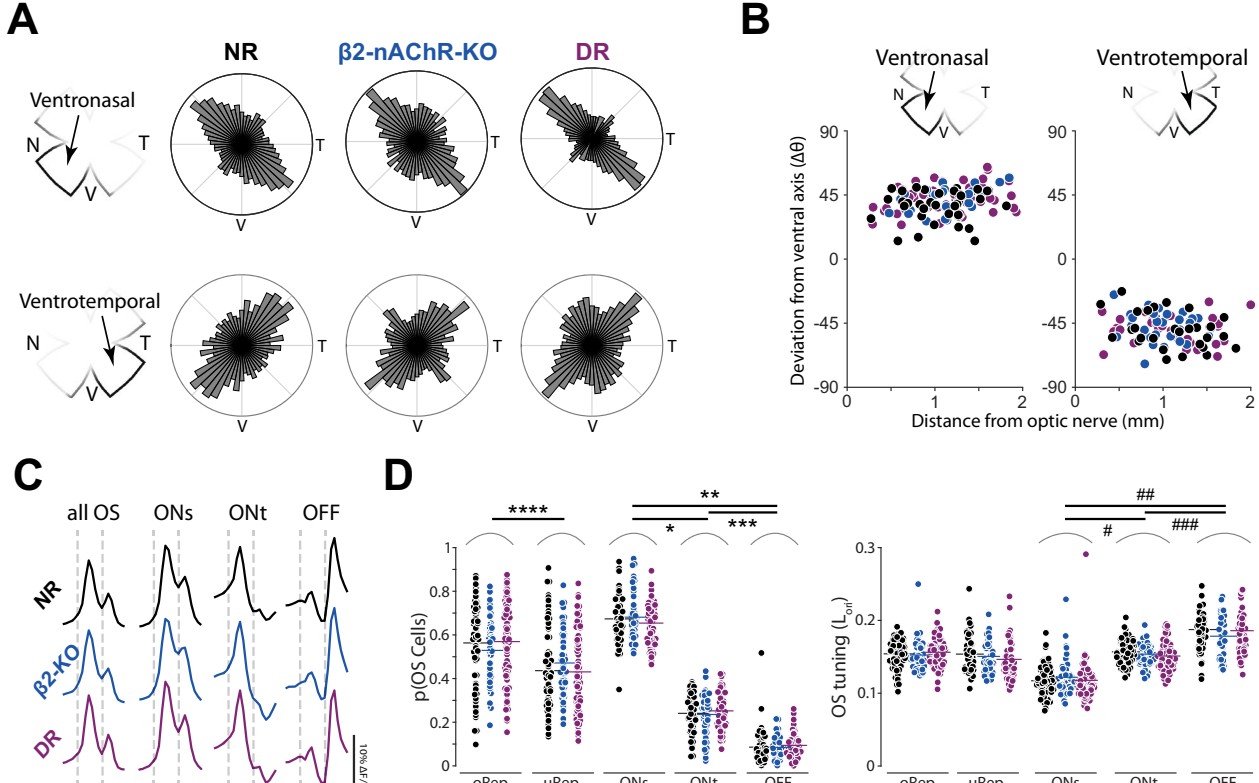

**Fig. 3 | Organization and tuning of retinal orientation selectivity are unaffected in mouse models of reduced retinal waves or visual deprivation. A** Polar histograms showing organization of Orientation Selectivity (OS) in ventronasal (top row) and ventrotemporal (bottom row) retina in normal reared (NR) mice, β2-nAChR-KO mice, and dark-reared (DR) mice. **B** Preferred orientation of the overrepresented orientations as a function of distance from the optic nerve in ventronasal (left) and ventrotemporal (right) retina across experimental conditions (n = 36 VN and 31 VT FOVs for NR, 27 VN and 28 VT FOVs for β2-KO, 46 VN and 35 VT FOVs for DR). **C** Average ΔF/F response to all moving bars for, from left to right, all OS, ONs, ONt, or OFF OS cells. **D** Proportion (left) and tuning (right) of OS subtypes based on their preferred orientation (oRep overrepresented orientation, uRep underrepresented orientation) or response to light onset/offset across experimental conditions. $N$ = 7760 OS cells for NR, 5464 OS cells for β2-KO, and 10,928 OS cells for DR. $*p = 4.3 \times 10^{-32}$, $**p = 4.5 \times 10^{-267}$, $***p = 5.8 \times 10^{-176}$, $****p = 5.5 \times 10^{-12}$, $^{\#}p = 6.1 \times 10^{-27}$, $^{\#\#}p = 6.6 \times 10^{-109}$, $^{\#\#\#}p = 8.4 \times 10^{-42}$. Three factor ANOVA followed by two-tailed Tukey test. Source data are provided as a Source Data file.

an ellipsoidal map centered in ventral retina is the best descriptor for the organization of orientation selectivity in the mouse retina.

The results of our model allowed us to make two predictions about the overrepresented preferred orientation. First, in both ventronasal and ventrotemporal retina, FOVs closer to ventral retina should exhibit an overrepresented preferred orientation that deviates more from the cardinal ventral axis than FOVs farther from ventral retina (Fig. 4D, left). Using the recorded coordinates of each FOV, we indeed found this prediction to be true, with the distribution of preferred orientations shifted more to the right (becoming more horizontal) in more ventral FOVs (Fig. 4D, right). We further subdivided FOVs depending on whether they were more proximal or more distal from the optic nerve and found greater differences between distributions in proximal FOVs than distal FOVs, which again aligns with our model (Fig. S8).

The second prediction of our model is that the overrepresented preferred orientation aligns with horizontal axes in central ventral retina and vertical axes in temporal retina. We analyzed a subset of mice where moving bars of light were presented in ventral and temporal retina (n = 938 orientation selective cells across 15 FOVs across 3 mice for ventral; n = 1891 orientation selective cells across 22 FOVs across 3 mice for temporal). Similar to ventronasal and ventrotemporal retina, we again observed a map that depicted one overrepresented preferred orientation (Fig. 4E). Unlike in ventronasal and ventrotemporal retina, the overrepresented preferred orientation aligned to horizontal axis in ventral retina, and to vertical axis in temporal retina, again confirming the predicted orientations of our model.

Lastly, we performed two more analyses to further validate our model. First, we used the retinal location of each orientation selective cell that belonged to the overrepresented group to build a vector flow field of the preferred orientation across the surface of ventral central retina, which again highlighted the abundance of horizontal orientation preference and matching the predicted angles of concentric ellipses in ventral retina (Fig. 4F). Second, in temporal retina, we analyzed FOVs that were either more dorsal or more ventral (±400 μm distance; Fig. 4G, left). We observed that the overrepresented preferred orientations deviated in a way that is predicted by our model (Fig. 4G, right). Together, these results indicate that orientation selectivity is mapped along concentric axes in the retina, similar to the superior colliculus[17,18].

## Discussion

In the present study, we made use of a rich dataset to thoroughly assess the organization of orientation selectivity in the mouse retina. In doing so, we made the following key discoveries. First, we found that in any given area of the mouse retina, one orientation is overrepresented, and this overrepresented orientation changes as a function of retinal location. Second, we demonstrated that there is an inverse relationship between the prevalence of orientation selective cell subtypes and their tuning properties. Third, we demonstrated that neural activity during development appears dispensable for the development of the organization of orientation selectivity. Lastly, we proposed and tested a model of how orientation selectivity is mapped across the surface of the retina.

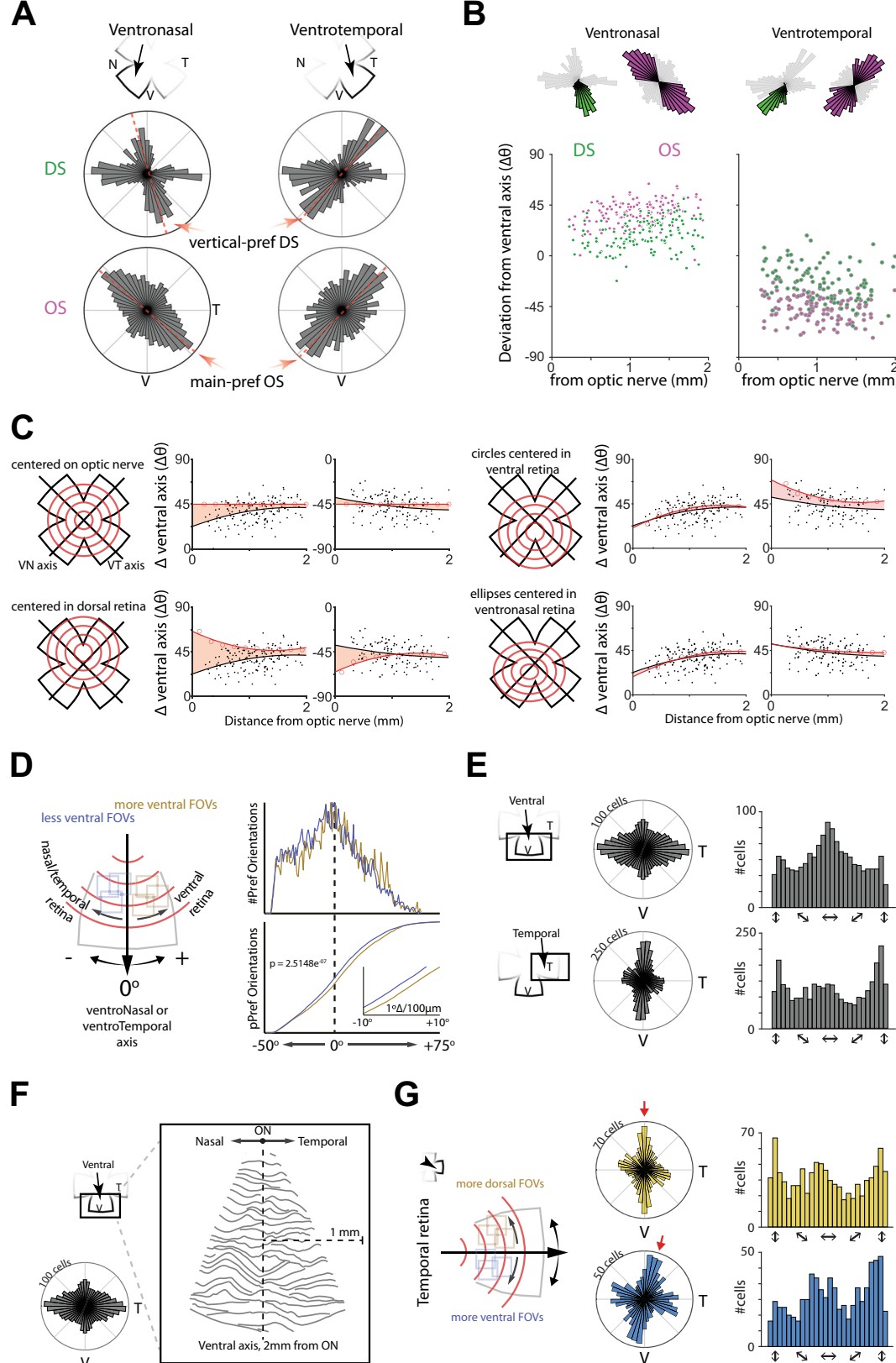

## Implications for how orientation selectivity is organized along the visual neuraxis

Our finding that orientation selectivity changes across the surface of the retina, and how it aligns along concentric ellipses centered in ventral retina, has implication for how orientation selectivity is organized downstream of the retina especially given the increase in published models of cortical and collicular organizations that factor in retinal organization and activity[25,37]. This is likely especially true in areas like the superior colliculus, which receives extensive retinal innervation and inherits some computations from the retina[38,39].

In the rodent superior colliculus, orientation selectivity changes as a function of location[19]. Specifically, there is evidence that

**Fig. 4 | The retinal orientation selectivity map is organized along concentric axes. A** Polar histograms showing organization of Direction Selectivity (DS, green) and Orientation Selectivity (OS, magenta) in ventronasal and ventrotemporal retina. **B** Angle deviation of the preferred direction for DS cells (green) and orientation for OS cells (magenta) from the ventral cardinal axis as a function of how far the FOV is from the optic nerve in ventronasal (left) and ventrotemporal (right) retina. **C** In silico modeling of how OS preferred axes map based on concentric circles anchored on the optic nerve (top left), dorsal retina (bottom left), and ventral retina (top right). The bottom right plot is a model of concentric ellipses anchored in ventronasal retina, which outputs predicted lines that deviate the least from the actual data. Red lines are the simulated preferred orientation axes, black data are the measured preferred orientations based on RGC location, and black lines are fits to experimental data. Red shading represents area between the two fits. **D** Left: schematic demonstrating analysis of preferred orientation of OS cells around the ventronasal and ventrotemporal axis to test predictions from the model in (**C**). FOVs closer to ventral retina (yellow) exhibit more positive-deviating preferred orientations from the ventronasal/ventrotemporal axis, whereas FOVs away from ventral retina (blue) exhibit more negative deviating preferred orientations from the ventronasal/ventrotemporal axis. Right: histogram and cumulative distribution of preferred orientations from more and less ventral FOVs. Less ventral FOVs exhibit statistically more negative deviating preferred orientations and statistically less positive deviating preferred orientations than more ventral FOVs. *KS test and permutation test reveal that the distributions are statistically significantly distinct. **E** Polar and linear histograms depicting distribution of preferred orientations in ventral (top) and temporal (bottom) retina. **F** Vector flow field of preferred orientations in ventral retina. **G** Polar and linear histograms depicting distribution of preferred orientations in temporal retina separated for fields of view that are either more dorsal (yellow) or more ventral (blue).

orientation selectivity in the superior colliculus is organized along concentric axes[17,18]. Importantly, retinal axons innervating the superior colliculus also exhibit the same organization properties as collicular neurons[17]. Because the superior colliculus receives precise anatomical and functional retinotopic projections from the retina[40,41], the fact that orientation selectivity changes as a function of location in retinal boutons in the superior colliculus aligns well with our presented findings. Some differences in the map of orientation selectivity in retinal neurons and retinal boutons may arise depending on whether retinal axons homogeneously project to the superior colliculus, which may not be the case in binocular zones[17].

In the rodent visual cortex, orientation selectivity is organized in a salt-and-pepper fashion[20,21]. Interestingly, orientation selectivity is initially organized in visual cortex[24]. It is possible that this initial organization reflects that of the retina during development, but this remains to be tested. Furthermore, assessing how perturbing the retinal orientation selectivity map affects the adult visual cortical map will reveal how much of cortical orientation selectivity is inherited from the retina.

### Functional implications of the orientation selectivity map

The fact that orientation selectivity exhibits a map across the surface of the retina is reminiscent of the discovery that direction selectivity also exhibits a map across the surface of the retina[3]. Both of these discoveries upend previously-thought notions that both direction and orientation selectivity map along location-independent cardinal axes. Our current understanding of the direction selectivity map suggests it aligns to the body and gravitational axis but can encode both translation and rotation information[3]. Direction selective cells are further divided into two major types: ON-OFF and ON direction selective cells. While ON direction selective cells are known to be important for the optokinetic reflex, no behavior has yet to be identified to specifically depend on ON-OFF direction selective cells. Critically, ON-OFF direction selective cells exhibit an overrepresented preference for forward motion[3,4], but whether this overrepresented preference matters for behavior is not yet known. In fact, the behavioral relevance of why direction selective cells are organized along axes of optic flow is not yet understood. Comparative studies in non-mammalian species suggest that the direction selective maps may reflect the degrees of freedom in animal movements[42]. Similar to direction selectivity, future studies will need to explore whether the map of the overrepresented orientation preference has a behavioral relevance and changes depending on anatomical position of the eye and motor behavior of the animal.

### Activity-dependent development of orientation maps likely occurs downstream of the retina

We show the development and organization of retinal orientation selectivity to be largely independent of neural activity. This was true for both spontaneous (i.e., retinal waves) and visually evoked activity. This seems to be distinct from direction selectivity, where retinal waves appear to be important for setting up horizontal direction selectivity[4,27] and where visual enrichment sharpens direction tuning[43].

There is conflicting evidence that neural activity is important for the organization of orientation selectivity downstream of the retina. For example, in the visual cortex, individual neurons exhibit strong selectivity at eye opening but the onset of visual experience both increases an individual neuron's tuning[44,45] and the connectivity between neurons sharing similar feature selectivities[28]. The organization of orientation selectivity initially exhibits an overrepresented orientation at eye opening but exhibits equal representation of all orientations by adulthood[24]. Visual experience is not necessary for this developmental process or the initial formation of cortical selectivity but abolishing both spontaneous and evoked activity does impair the formation of the adult orientation selectivity map[24]. Other studies show that stripe-rearing increases overrepresentation of experienced orientation in layer 2/3 but not in deeper layers[46]. In higher visual areas, the tuning of orientation selective cells continues to develop following the onset of visual experience at eye opening[47]. Lastly, cortical orientation selectivity seems spared in a mouse model of reduced cholinergic waves[27].

Although the past decade has seen tremendous research on understanding the superior colliculus[48], much less is known about how orientation selectivity develops compared to visual cortex. This is in part because the exact organization of adult orientation selectivity (and direction selectivity) in the superior colliculus is still controversial. Nonetheless, there is some evidence that activity plays a role in collicular orientation selectivity. Mouse models that lack cholinergic waves exhibit a drastic reduction in both direction and orientation selective cells, specifically those that would respond to horizontal motion[27]. Dark rearing on the other hand, seems to only have limited effects on the development of collicular orientation selectivity[49]. Thus, given our results, we think that the wave-dependent effects on the superior colliculus likely occurs at the superior colliculus, consistent with previous reports[25].

In conclusion, we used a bulk approach to identify the functional retinal orientation/axial selectivity map. We believe this could only be achieved by analyzing thousands of orientation selective retinal neurons through various retinal locations. That said, the retina has become an excellent model system for comparing functional subtypes[10] with transcriptomic subtypes[50,51]. Thus, an important outstanding future goal will be to combine techniques to ascertain how much the functional distinctions match with transcriptomic distinctions. Additionally, demonstrating the orientation selectivity map may also help identify potential candidate molecular gradients that could be involved in its formation.

## Methods
### Ethics
All animal procedures were approved by the UC Berkeley Institutional Animal Care and Use Committee, as well as the Vanderbilt Animal Care

and Use Program and conformed to the NIH Guide for the Care and Use of Laboratory Animals, the Public Health Service Policy, and the SFN Policy on the Use of Animals in Neuroscience Research.

## Animals

All mice in this study were aged between 30–60 days and of both sexes and (except for the visual deprivation group) were housed in a 12/h day/night cycle vivarium. All experiments are done on C57BL/6J mice (Jackson stock: 000664). To study the role of spontaneous activity, we used the β2-nAChR-KO mice where the beta subunit of the nicotinic acetylcholine receptor is knocked out. To study the role of visual experience, we used mice that were born and raised in rooms that either had 12-h day/night cycle or 24 h darkness. For the dark-reared mice, all animal husbandry was conducted with red light, which minimizes stimulation of photoreceptors. Deprivation of light in the dark-reared mice was confirmed using a custom-built light meter attached to cages that logged data at 1 Hz over the course of a month (https://github.com/Llamero/Light_Color_and_Intensity_Datalogger).

To study how much of the orientation selectivity map comes from retinal ganglion cells, we used transgenic mice where GCaMP6s (Jackson stock: 028866) was expressed under the vgtlut2cre promoter (Jackson stock: 028863). These mice are also in the C57Bl/6J genetic background.

## Dataset

We used the dataset that was collected to study the direction selectivity map in ref. 4. Please refer to the "Data acquisition details" section of that publication for information on how the data were acquired.

## Data analysis details

**Image processing.** Raw movies were motion-corrected and normalized into $\Delta F/F_0$ automatically using a custom-made FIJI (ImageJ 1.54F) macro that was run in ImageJ v1.52n (macro name in project's GitHub folder: RegisterAndCalcDFOF_withGFP_v2.ijm). Briefly, (1) movies were motion corrected using the "correct 3D drift" plugin in FIJI on a duplicate of the raw data that had been averaged in the time dimension (zMean = 30 s). (2) Frames where the light stimulus occurred were removed to isolate baseline $F$. (3) The baseline $F$ was subtracted from the raw $F$ movie, and this result was divided by the baseline $F$. The resulting $\Delta F/F_0$ movies were then transferred to MATLAB R2022a for further image analysis.

**Semiautomated cell segmentation.** Retinal ganglion cells (RGC) from each field of view (FOV) were completely annotated using a combination of automated and manual segmentation. For automation, we utilized the open-source segmentation tool Cellpose 2.0[29,52]. Cellpose uses machine learning to train neural networks for cell segmentation. Videos from each FOV were processed for motion correction then converted to an average intensity projection to clearly show each RGC. Ten example FOV projections were then randomly selected to train the Cellpose model to our dataset. The resulting model was subsequently used to segment all other FOV projections. Cellpose accurately identified the majority of cells in each FOV but was typically incomplete. To ensure full annotation of each FOV, Cellpose-generated masks were then opened in ImageJ and converted to region of interests (ROI) using the "label image to ROI" function in the BIOP plugin. Annotation of each FOV was then completed via manual tracing using the same average intensity projection as above, creating full ROI sets for each FOV. Completed ROI sets were then utilized for downstream computational query.

**Response quality index.** We used the following signal-to-noise calculation to determine how consistently a cell responded to the presentation of the bar stimulus:

$$QI = \frac{Var[\langle C \rangle r]t}{\langle Var[C]t \rangle r} \tag{1}$$

where $C$ is the T (time samples) by R (stimulus repetitions) response matrix and $\langle C \rangle r$ and $Var[C]t$ denote the mean and variance across the $r$ (stim rep) and $t$ (time samples) dimensions. QI is equal to 1 if all responses to the same stimulus are exactly the same (and thus the mean represents consistent responses). We used a threshold of QI > 0.6 based on previous work.

**Identification of direction and orientation selective cells.** Direction selectivity was calculated as was done on the original dataset[4]. Specifically, for every neuron, we computed the average peak $\Delta F/F_0$ response for every stimulus direction and computed the vector sum (VS). Using the direction of the vector sum, we identified each cell's preferred direction. We then used two methods to calculate direction selectivity:

$$L_{dir} = \left| \frac{\sum_k R(\theta_k)e^{i\theta_k}}{\sum_k R(\theta_k)} \right| \tag{2}$$

Where $R(\theta_k)$ is the response to angle $\theta_k$. We also computed the direction selectivity index:

$$\begin{aligned}\text{Direction selectivity index} = (\Delta F/F_0 \text{pref} - \Delta F/F_0 \text{null})/ \\ (\Delta F/F_0 \text{pref} + \Delta F/F_0 \text{null})\end{aligned} \tag{3}$$

Where "pref" is the direction angle closest to the vector sum's direction and "null" is 180° rotated from pref.

We used the average peak $\Delta F/F_0$ response for every stimulus direction to calculate the orientation selectivity of each cell. Specifically, we calculated the vector direction and length in orientation space as described in ref. 30:

$$L_{ori} = \left| \frac{\sum_k R(\theta_k)e^{2i\theta_k}}{\sum_k R(\theta_k)} \right| \tag{4}$$

where $R(\theta_k)$ is the response to angle $\theta_k$. $L_{ori}$ provides the strength of orientation tuning, whereas the phase of the complex component provides the preferred orientation. We also computed the orientation selectivity index by first averaging the peak response of directions that were 180° to one another to obtain 4 orientations. Then we used the following formula to calculate the orientation selectivity index:

$$\begin{aligned}\text{Orientation selectivity index} = (\Delta F/F_0 \text{pref} - \Delta F/F_0 \text{orthogonal})/ \\ (\Delta F/F_0 \text{pref} + \Delta F/F_0 \text{orthogonal})\end{aligned} \tag{5}$$

where "pref" is the orientation that exhibited the largest peak response and "orthogonal" is the orientation orthogonal to the preferred orientation.

**Statistical determination of direction and orientation selectivity.** The following statistical approach was used to determine which cells were significantly direction or orientation selective: For each cell, the $L_{dir}$ and $L_{ori}$ was first calculated. Then, for 1000 permutations in silico, the directions of the moving bar stimuli were block-shuffled and the $L_{dir}$ and $L_{ori}$ were again calculated. The cell's $L_{dir}$ and $L_{ori}$ calculated from the non-permuted dataset was ranked against all the $L_{dir}$ and $L_{ori}$ calculated from the permuted dataset, respectively. If the cell's actual $L_{dir}$ or $L_{ori}$ ranked higher than 95% of the permuted $L_{dir}$ or $L_{ori}$, it was determined to be significantly direction or orientation selective, respectively.

**Clustering analysis based on ON-OFF response or preferred orientation.** We used a clustering method to discern how many functional groups our data best fit into for both the OS cells' ON-OFF response and preferred orientation[4,53]. For each measure, we used a combination of iterating K-means clustering with increasing number of groups followed by a silhouette analysis. This method optimizes the set of clusters with respect to the distance between each point and the centroid of its cluster, summed for all points. We compared 2–10 cluster numbers, and we calculated the fitness of clustering by using the silhouette value (SV):

$$SV(i) = (b(i) - a(i))/ \max(a(i), b(i)) \tag{6}$$

where $a(i)$ is the average distance between $i$ and all other data within the same cluster (called measure of cohesion), and $b(i)$ is the average distance between $i$ and all points in the nearest cluster (called measure of separation from the closest other cluster). A SV close to 1 indicates data perfectly clustered, whereas a SV close to 0 reflects data which are ambiguously clustered. We repeated this analysis on permuted data to gain a better understanding of what this silhouette analysis outputs on a randomized dataset.

**Statistical tests to determine differences in proportion or tuning of cells.** We used either a two factor (subtype × direction; Fig. 1) or three factor (subtype × direction × condition; Fig. 3) ANOVA. If main effects were significant, we followed up with a Tukey test to compare individual groups. Interaction was never found to be significant. All statistics were performed in MATLAB.

**Modeling the slope of concentric circles and ellipses on the ventronasal and ventrotemporal axis.** We created a user-friendly app in MATLAB R2022a that allows the user to set the parameters of the center of the concentric circles or turn the circles into ellipses. The app is available on our Github page: https://github.com/TiriacLabCodeShare/OSproject.

For the concentric circles, the app utilizes a built-in MATLAB function, called "linecirc", that identifies the intersection of a given line and circle. We then used this function to identify the locations on a circle where ventronasal (VN) and ventrotemporal (VT) axes would cross a circle with center $x_o, y_o$. Next, we traced the intersections of the VN and VT axes back to the center of the concentric circles to obtain the slope of that line. Next, we calculated the slope of the orthogonal line (which results in the slope of a line tangential to the point intersected by either VN or VT) and calculated the predicted angle of this tangential slope (or vector). We then repeated this process iteratively for circles of increasing diameter to obtain the red modeled lines seen in Fig. 4C.

For the concentric ellipses, we used an open-source function "lineEllipse" (KSSV (2023). lineEllipse (https://www.mathworks.com/matlabcentral/fileexchange/68619-lineellipse), MATLAB Central File Exchange) that again identifies the intersection of an ellipse with major axis a, minor axis b, and center O, with two different lines (VN and VT). We next used the following formula to identify the slope of the line tangential to the intersected point:

$$Slope_{VNtangent} = -(y_{radius}^{2*}(VN_{intercept} - X_{center}))/ \tag{7}$$
$$(X_{radius}^{2*}(VN_{intercept} - y_{center}))$$

$$Slope_{VTtangent} = -(y_{radius}^{2*}(VT_{intercept} - X_{center}))/ \tag{8}$$
$$(X_{radius}^{2*}(VT_{intercept} - y_{center}))$$

As with the concentric circles, we calculated the predicted angle of this tangential slope (or vector).

We fit quadratics to both the actual and modeled datasets and computed the area between the lines (red shadings in Fig. 4C).

## Reporting summary

Further information on research design is available in the Nature Portfolio Reporting Summary linked to this article.

## Data availability

Processed data are available on our lab's Github page: https://github.com/TiriacLabCodeShare/OSproject. The non-processed dataset is available upon request. Source data are provided with this paper.

## Code availability

Analysis code is available on our lab's Github page: https://github.com/TiriacLabCodeShare/OSproject. Code also deposited on Zenodo, https://doi.org/10.5281/zenodo.11236270.

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

## Acknowledgements

We thank Dr. Marla Feller for her helpful comments on earlier versions of the manuscript. A.T. was supported by NIH R00EY030909.

## Author contributions

D.V., F.O., N.S., N.C. and A.T. carried out investigation, analysis, and edited the manuscript. D.V. and A.T. conceptualized the study, wrote software to analyze dataset, and wrote the original draft. A.T. acquired funding and supervised the study.

## Competing interests

The authors declare no competing interests.
