## [Peer Review File · Nature Communications]

Development and organization of the retinal orientation selectivity mapREVIEWER COMMENTS

Reviewer #1 (Remarks to the Author):

This manuscript describes a series of experiments and modeling aimed to understand the spatial organization and development of orientation selectivity (OS) map in mouse retina. While we have a detailed understanding of the mechanisms involved in formation and organization of direction selectivity (DS) map, our knowledge of OS mechanisms in retina remains immature.

The authors attempt to bridge this gap using two photon calcium imaging from a large population of neurons in ganglion cell layer in response to moving bar stimuli. OS cells reported in this study fall broadly into three categories based on response polarity and duration: ON sustained (ONs), ON transient (ONt) and OFF. Through detailed analysis, the authors demonstrate that ONs cells are weakly tuned but more prevalent whereas the opposite holds true for OFF cells. Neither the proportion nor the tuning strength of different OS subtypes were altered in mice lacking cholinergic or glutamatergic retinal waves during development.

In what is perhaps the most novel component, the authors demonstrate that the preferred orientation of OS cells pooled across subtypes changed as a function of retinal location. Further analysis and *in silico* modeling, enable the authors to determine that the OS map is an ellipsoid centered in the ventral retina.

I have no technical problems with the experiments but my major concern with the paper is the way that it is presented to readers. The impressive amount of data, analysis and modeling are shadowed by the order of figures. The fact that there's a spatial dependence of OS tuning is striking and should be presented first. Development of OS map is of subsequent concern and should be discussed later in the paper. Also, the lack of clear introduction, results and discussion sections makes it a difficult read. Regardless, the findings in this paper will appeal to a broader audience outside retina specialists given the ubiquitous nature of OS across the visual system.

Specific comments:

1. OSI and DSI are calculated differently. It would be good if the authors adhere to a vector sum for both cases. The authors cite Mazurek et al. 2014 and they should probably use Lori and Ldir used in that study.
2. Figure 1C: Why do cells with higher Lori have high DSI? Are these cells both OS and DS (seems ambiguous)?
3. What is the cutoff threshold for a cell to be considered OS/DS? The authors perform statistical tests based on random shuffling of a cell's response to different moving bar directions to classify cells as OS/DS. While this is a fine way of determining OS/DS, it compares a cell's response to only itself across multiple iterations and not to the entire population of neurons in the dataset. It would be good to see a histogram of cells for Lori and DSI.
4. Line 60-61: It would be good to see rose plots of preferred orientations for the 3 OS types to support this claim. The data is there in Fig. 2A but not separated by OS types. Also the lack of a scale bar makes it difficult to draw conclusions about these histograms.
5. Fig 1G, 1H: Direction for OS cells doesn't make much sense. It should be replaced with orientation. Similarly in S1C there should be two orientations instead of four.
6. One major caveat of this study is the lack of knowledge about whether the imaged cells are RGCs or ACs. Combined data across subtypes is used in Figures 2 and 3. Are these groups all OS RGCs? They probably represent a mixed population of RGCs and displaced ACs. In that case is the OS map an RGC map? If not, then this map doesn't affect downstream visual regions as the authors mention in lines 135-140 and these claims should be softened. Equivalent claims can be made in case of DS map(s) because it has been studied in genetic lines labeling specific DS RGC types. Conducting these imaging studies in a RBPMS-Cre line, would have enabled the authors to

reach much more significant conclusions about how an OS map in retina affects downstream visual regions.

Minor:

1. I like the minimalistic approach but the time axis should be labeled in Fig 1E,F.
2. Fig 1F: The authors should label that these cells are in the ventronasal retina.
3. A better abbreviation for optic nerve (ON) is needed since it can be confused with response polarity of cells.
4. Line 102: axes is spelled wrong.

Reviewer #2 (Remarks to the Author):

Vita et al. analyze previously published 2-photon calcium imaging datasets of retinal ganglion cell (RGC) responses to explore potential maps of orientation selectivity across the mouse retina and probe the mechanisms of their development. The authors suggest that orientation selectivity changes across the surface of the retina, aligning with concentric ellipses centered in the ventral retina. Although this is an exciting and creative idea, and the figures and text are well-structured and accessible, I do not think the data support this conclusion. I outline why and suggest potential ways to overcome this in the following specific comments.

Specific comments

1.) Four orientation-selective RGC types have been identified in the mouse retina (Nath and Schwartz JNeurosci 2016, Nath and Schwartz Nat Commun 2017). These differ in their contrast (ON vs. OFF) and orientation (horizontal vs. vertical) preferences and have distinct dendritic morphologies and transcriptomic profiles (Goetz et al. Cell Rep 2022). In functional studies, the orientation selectivity indices (OSIs) of >90% of the ON and OFF OS RGC types were >0.2 (Nath and Schwartz JNeurosci 2016, Nath and Schwartz Nat Commun 2017). By contrast, the vast majority of cells considered to be orientation selective in the present study have OSIs <0.2. Thus, it is unclear whether the maps revealed by Vita et al. apply to the OS RGC types. To test whether they do, the authors need to identify OS RGCs by their OS responses plus an independent parameter. Goetz et al. (2022) identified marker genes that could be labeled by post-hoc in situ hybridization after 2-photon calcium imaging to determine which responses belong to the OS RGC types. It is worth noting that Nath and Schwartz (2016) mapped the positions and tuning of ON OS RGC types across the retina and observed no region-dependent changes in the orientation preferences.

2.) For the reasons given in the previous point, the asymmetries observed by Vita et al. likely, at least in part, reflect differences in the orientation preferences of non-OS RGC types. This could still be interesting, but I am concerned that the observed maps could be an artifact of the transition from the retina in vivo as a hemisphere to a flattened preparation with relieving cuts. To overcome these concerns, one or both of the following would be needed:

2a.) The authors suggest that orientation preferences are aligned with concentric ellipses centered in the ventral retina. According to this model, the variation in orientation preferences in the dorsal retina should deviate from that in the ventral retina (where the ellipses are centered) in a predictable way. This prediction should be tested in recordings from the dorsal retina.

2b.) Alternatively, and arguably most convincing, would be to show that these orientation maps exist in vivo. This could be tested either by two-photon imaging or electrophysiological recordings of RGC axons in retinorecipient targets. Some datasets that may allow such analyses have already been published (e.g., Liang et al. Cell 2018, Sibille et al. Nat Commun 2022).

Minor comment:

1.) In Figure 2D, please sort the neurons within each type by their orientation preferences.

Reviewer #3 (Remarks to the Author):

In this manuscript, the authors investigate the orientation selectivity (OS) of retinal ganglion cells (RGCs) using an existing dataset originally generated by Tiriach et al. in 2022. They identify three functional OS RGC subtypes, with the rare OFF-OS cells exhibiting the highest selectivity among them. The study analyzes data from normally reared, dark-reared, and genetically retinal wave-affected mice, revealing robust OS development across all conditions. Additionally, the authors find differences between the global maps of OS and direction selectivity (DS).

While this manuscript presents a well-executed study on RGC orientation selectivity and provides valuable insights into RGC information processing, I have some concerns regarding its suitability for publication in a journal with a broad readership, at least with its current form. The manuscript appears somewhat incremental, with limited conceptual advancement beyond the authors' 2022 paper, the work by Baden et al. in 2016, and etc. Thus, it requires substantial revisions and expansion to enhance its significance and clarity for a broader readership. Here are my recommendations:

1. Conceptual Advancement and Discussion:

- The manuscript, apparently initially prepared as a brief report, lacks substantial expansion and discussion. The authors should delve into the biological advantages of having an overrepresented OS axis slightly offset from the DS axis. Furthermore, the implications of OFF-OS cells being more selective than ON-OS cells should be thoroughly discussed.
- The authors should consider the broader context of cortical OS/DS development, including previous work on experience-independent OS development in the visual cortex.
- While stating "implications for understanding how OS is organised downstream of the retina" (line 24), the authors omitted several crucial prior studies on experience-independent OS development in the visual cortex. For instance, Ko, Mrsic-Flogel et al. (2013), Hagihara, Ohki et al. (2015), and Hillier, Roska et al. (2017) have made significant contributions in this regard. Particularly, Hillier et al. extensively discussed the substantial influence of retinal DS on cortical DS/OS. It is essential for the authors to thoroughly examine their findings and their broader implications within the context of cortical OS/DS, encompassing not only colliculus-related research but also these noteworthy cortical studies.
- The retina has proven to be an excellent model for molecular and transcriptomic studies focused on specific cell types, and among these, DS cells have exemplified the convergence of molecular identity and function remarkably (Kim, Sanes et al. 2008; Yonehara, Roska et al. 2009, etc). It would be highly beneficial for future research to include a discussion on OS cells, particularly OFF cells, as they present an avenue for further exploration in this context.

2. Methodological Clarity:

The methods section could benefit from greater clarity. Specific details, such as the time-windows used for quantifying visual responses and the choice between peak dF/F and area-under-curve for responses, should be clearly outlined. Robustness against parameter selections should be demonstrated.

3. Inclusion of OS-DS Population:

Throughout the study, the authors treat OS cells and DS cells as mutually exclusive populations, a standard approach in the field for functional cell type classification. However, in terms of information representation and processing, disregarding the orientation selectivity of DS cells can be misleading. Indeed, some DS cells exhibit highly-tuned orientation selectivity (Figure 1C). The authors should investigate whether including this OS-and-DS population in the analysis of OS cells leads to different outcomes.

Also, the manuscript lacks clarity regarding data inclusion criteria. If OS and DS cells are separately used to produce maps in analyses for Figure 3, the authors should explore the effect of

including the OS-and-DS population cells or all visually responsive cells on OS/DS map axes analysis.

4. Neuropil Contamination:

Considering that calcium imaging was conducted using chemical AM dyes, it is expected that the dataset may include considerable neuropil contamination owing to tissue scattering. Such contamination can potentially diminish orientation selectivity (OS) by adding non-selective signals to actual selective signals. Given the sparse presence of OFF responsive cells, it is plausible that OFF cells are less susceptible to this signal contamination when compared to ON OS cells. To validate that the tuned OS observed in OFF cells is not simply an artefact, the authors should systematically vary the degree of neuropil signal subtraction, as demonstrated in studies such as Kerlin, Reid et al. (2010) and Hagihara, Ohki et al. (2015).

Other minor points:

1. The title of Figure 1 is somewhat indirect. Consider a more direct description to convey the greater tuning of OFF OS cells.
2. In line 102, "axed"
3. In Figure 1, please clarify the correspondence between B and A.
4. Consider including pixel-based orientation/direction color-map examples in Figure 3 to visually illustrate the extent of overrepresentation.
5. In Figure 1, for panels G and H, "direction" should be replaced with "orientation."

Reviewer #1 (Remarks to the Author):

This manuscript describes a series of experiments and modeling aimed to understand the spatial organization and development of orientation selectivity (OS) map in mouse retina. While we have a detailed understanding of the mechanisms involved in formation and organization of direction selectivity (DS) map, our knowledge of OS mechanisms in retina remains immature.

The authors attempt to bridge this gap using two photon calcium imaging from a large population of neurons in ganglion cell layer in response to moving bar stimuli. OS cells reported in this study fall broadly into three categories based on response polarity and duration: ON sustained (ONs), ON transient (ONt) and OFF. Through detailed analysis, the authors demonstrate that ONs cells are weakly tuned but more prevalent whereas the opposite holds true for OFF cells. Neither the proportion nor the tuning strength of different OS subtypes were altered in mice lacking cholinergic or glutamatergic retinal waves during development.

In what is perhaps the most novel component, the authors demonstrate that the preferred orientation of OS cells pooled across subtypes changed as a function of retinal location. Further analysis and in silico modeling, enable the authors to determine that the OS map is an ellipsoid centered in the ventral retina.

I have no technical problems with the experiments but my major concern with the paper is the way that it is presented to readers. The impressive amount of data, analysis and modeling are shadowed by the order of figures. The fact that there's a spatial dependence of OS tuning is striking and should be presented first. Development of OS map is of subsequent concern and should be discussed later in the paper. Also, the lack of clear introduction, results and discussion sections makes it a difficult read. Regardless, the findings in this paper will appeal to a broader audience outside retina specialists given the ubiquitous nature of OS across the visual system.

We thank the reviewer for the comments. To improve the readability of the manuscript, we have greatly revised the introduction, results, and discussion, as well as the order that the data are presented (the map now comes before the developmental story). More specific changes in responses below.

Specific comments:

1. OSI and DSI are calculated differently. It would be good if the authors adhere to a vector sum for both cases. The authors cite Mazurek et al. 2014 and they should probably use L_{ori} and L_{dir} used in that study.

As requested, and because of the work of Mazurek et al., 2014, we now use both L_{ori} and L_{dir} . Since OSI and DSI calculations are common in this field, we added a supplementary figure (Figure S1 & S5) that compares the data using the OSI and DSI calculation.

2. Figure 1C: Why do cells with higher L_{ori} have high DSI? Are these cells both OS and DS (seems ambiguous)?

The issue with Figure 1C as we used to have it presented is that density information was lost. We modified the figure to be a heatmap (Figure 1C) and also added a supplemental figure that delves deeper into this analysis. Cells with both high orientation and direction tuning that pass the statistical tests for orientation and direction selectivity are rare (<1%).

Separately, we calculated how many of our OS cells also exhibit statistically significant DS and vice versa. Our results show that 3% of OS cells are also statistically significantly DS, and 8% of DS cells are also statistically significantly OS. We also added an analysis where we remove these DS & OS cells but this does not affect the results (Figure S5).

3. What is the cutoff threshold for a cell to be considered OS/DS? The authors perform statistical tests based on random shuffling of a cell's response to different moving bar directions to classify cells as OS/DS. While this is a fine way of determining OS/DS, it compares a cell's response to only itself across multiple iterations and not to the entire population of neurons in the dataset. It would be good to see a histogram of cells for Lori and DSI.

Indeed, we used a random shuffling and a 95% confidence interval to determine whether cells are statistically significantly DS or OS. As requested, we added a new analysis in Figure S1 and pasted below for convenience that demonstrates the DS and OS tuning of cells that our analysis classifies as either DS or OS (or for the rest of the non OS or DS cell population). For example, the left most column shows Lori values for the OS cells (top), DS cells (middle), and non-OS or DS cells (bottom).

4. Line 60-61: It would be good to see rose plots of preferred orientations for the 3 OS types to support this claim. The data is there in Fig. 2A but not separated by OS types. Also the lack of a scale bar makes it difficult to draw conclusions about these histograms.

We added a new figure 2 that shows the rose plots and linear histograms of preferred orientations for the 3 OS types as well as their mapping across ventral retina. We also added scale bars to all of our rose plots. Lastly, to increase readability, we added linear histograms to many of our rose plots as they help with comparing different directions/orientations.

5. Fig 1G, 1H: Direction for OS cells doesn't make much sense. It should be replaced with orientation. Similarly in S1C there should be two orientations instead of four.

The word "direction" in Figure 1G and H now correctly says orientation and we fixed figure S2C (which used to be Figure S1C).

6. One major caveat of this study is the lack of knowledge about whether the imaged cells are RGCs or ACs. Combined data across subtypes is used in Figures 2 and 3. Are these groups all OS RGCs? They probably represent a mixed population of RGCs and displaced ACs. In that case is the OS map an RGC map? If not, then this map doesn't affect downstream visual regions as the authors mention in lines 135-140 and these claims should be softened. Equivalent claims can be made in case of DS map(s) because it has been studied in genetic lines labeling specific DS RGC types. Conducting these imaging studies in a RBPMS-Cre line, would have enabled the authors to reach much more significant conclusions about how an OS map in retina affects downstream visual regions.

This is a valid point as we do indeed think there are ACs present in the calcium dye dataset, especially since OS ACs have been described. However, we do not think this affects our conclusions that the map that we identify has implications for OS maps downstream of the retina for the following two reasons.

First, per imaging FOV, all three OS subtypes demonstrate the same overrepresented orientation preference (as can be seen in the new Figure 2). Thus, unless displaced ACs overwhelmingly contaminate all three OS functional groups that we detect, the more likely scenario is that ACs and RGCs exhibit the same OS map.

Second, we analyzed data from a cohort of *Vglut2cre:Gcamp6s* mice. In the RGC layer, *vglut2* is present only in RGCs and not in amacrine cells (thus this is similar to the experiment the reviewer proposed with the RBPMS-cre line). This mouse line confirmed several aspects of the calcium dye dataset:

- 1) We detected roughly the same number of OS cells per imaging field of view.
- 2) Even in this dataset, we could obtain the same 3 functional subtypes: ONs, ONt, and OFFt.
- 3) Like in the calcium dye dataset, each imaging FOV contains an orientation that is overrepresented and this overrepresentation is shared across the 3 OS subtypes.

The results of the *vglut2-cre:Gcamp6s* experiments are in the new Figure S6 (and pasted below for convenience).

Minor:

1. I like the minimalistic approach but the time axis should be labeled in Fig 1E,F.

We labelled the time axis in all plots

2. Fig 1F: The authors should label that these cells are in the ventronasal retina.

Added information in the caption and panel 1F

3. A better abbreviation for optic nerve (ON) is needed since it can be confused with response polarity of cells.

We stopped using ON as an abbreviation for optic nerve.

4. Line 102: axes is spelled wrong.

This is now fixed

Reviewer #2 (Remarks to the Author):

Vita et al. analyze previously published 2-photon calcium imaging datasets of retinal ganglion cell (RGC) responses to explore potential maps of orientation selectivity across the mouse retina and probe the mechanisms of their development. The authors suggest that orientation selectivity changes across the surface of the retina, aligning with concentric ellipses centered in the ventral retina. Although this is an exciting and creative idea, and the figures and text are well-structured and accessible, I do not think the data support this conclusion. I outline why and suggest potential ways to overcome this in the following specific comments.

Specific comments

1.) Four orientation-selective RGC types have been identified in the mouse retina (Nath and Schwartz JNeurosci 2016, Nath and Schwartz Nat Commun 2017). These differ in their contrast (ON vs. OFF) and orientation (horizontal vs. vertical) preferences and have distinct dendritic morphologies and transcriptomic profiles (Goetz et al. Cell Rep 2022). In functional studies, the orientation selectivity indices (OSIs) of >90% of the ON and OFF OS RGC types were >0.2 (Nath and Schwartz JNeurosci 2016, Nath and Schwartz Nat Commun 2017). By contrast, the vast majority of cells considered to be orientation selective in the present study have OSIs <0.2. Thus, it is unclear whether the maps revealed by Vita et al. apply to the OS RGC types.

There are a few ways we would like to address this comment.

First, we ran an analysis of the mapping of ventronasal and ventrotemporal OS cells if we only used cells that exhibited an OSI > 0.2, similar to the Nath and Schwartz studies (new Figure S5 and here at left for convenience). The maps still show the same general features, namely an overrepresentation of preferred orientation that changes depending on retinal location. In the DS field, several studies have raised questions about how to pick a DSI threshold since there is no natural breakpoint between non DS cells and DS cells. Instead, these papers have developed novel threshold independent test that rely on permutation tests to identify statistically significant DS cells (Baden et al, 2016, Sabbah et al, 2018, Tiriac

et al, 2022). We chose to do the same here for OS since we believe this analysis more accurately reflects the functional OS map in the retina. Regardless of the criteria used to define an OS cell, the identified OS map remained the same (Figure S5).

Second, in our study, we calculated both OSI and L_{ori} values but only displayed L_{ori} , which will be different from OSI values. We chose to use L_{ori} because of the arguments described in Mazurek et al., 2014, namely that the OSI calculation is more prone to tabulate higher OSI values than the true OS of cells (Figure 2 in Mazurek paper). We added a new supplementary figure (Figure S1) that compares OSI and L_{ori} values in statistically-significantly OS and non OS cells. Importantly, notice that our analysis does a good job of sequestering OS cells from DS cells or the rest of the non-OS population using either OSI or L_{ori} values.

Lastly, the Nath and Schwartz JNeurosci 2016, Nath and Schwartz Nat Commun 2017 are both studies using single-cell electrophysiology, which at least in the DS field, results in higher DSIs (We expect the same would be true for OSI) than calcium imaging studies. Our OSI values are comparable to those found in the Baden et al., *Nature*, 2016 calcium imaging study.

Nonetheless, as we showed in figure 1H, the results of our permutation test to statistically identify OS cells does a great job of identifying cells with much higher L_{ori} values than non-OS cells, so we feel confident that these cells are orientation selective cells.

To test whether they do, the authors need to identify OS RGCs by their OS responses plus an independent parameter. Goetz et al. (2022) identified marker genes that could be labeled by post-hoc in situ hybridization after 2-photon calcium imaging to determine which responses belong to the OS RGC types.

The intent of our study is to broadly highlight the functional retinal OS map, which would include amacrine cells and retinal ganglion cells. We believe that the only way to obtain this map was to collect high-throughput data. That said, and as asked by reviewer 1, we see the value of specifically investigating the OS RGC map. Thus, we analyzed a dataset from mice where Gcamp is expressed only in RGCs (below copied from response to reviewer 1).

... we analyzed data from a cohort of Vglut2cre:Gcamp6s mice. In the RGC layer, vlgut2 is present only in RGCs and not in amacrine cells (thus this is similar to the experiment the reviewer proposed with the RBPMS-cre line). This mouse line confirmed several aspects of the calcium dye dataset:

- 1) We detected roughly the same number of OS cells per imaging field of view.*
- 2) Even in this dataset, we could obtain the same 3 functional subtypes: ONs, ONt, and OFFt.*
- 3) Like in the calcium dye dataset, OFFt cells are more tuned than ONs cells. ONt cells seem to be as tuned as OFFt cells in the gcamp dataset.*
- 4) Like in the calcium dye dataset, each imaging FOV contains an orientation that is overrepresented and this overrepresentation is shared across the 3 OS subtypes.*

We view this current study as the initial discovery of the OS retinal map, which includes both amacrine cells and RGCs. Some of our additional experiments (Gcamp in RGCs) now also demonstrate that RGCs are largely represented in this map. Future work from our lab will delve into individual RGC types. To this point, reviewer 3 also commented that work on individual subtypes, using findings from transcription studies, is a fascinating future avenue, but only asked us to discuss it, highlighting that although a clear follow up, is beyond the scope of this study. Lastly, we believe that reporting on the functional retinal OS map now is critical given a recent study that demonstrates an OS maps in retinal boutons in the SC (Malmazet et. al, 2024).

It is worth noting that Nath and Schwartz (2016) mapped the positions and tuning of ON OS RGC types across the retina and observed no region-dependent changes in the orientation preferences.

The reviewer is referring to this figure from Nath and Schwarz, 2016 (pasted below). The first point to make is that even in this study, one can observe slight deviations in the preferred orientations throughout the surface of the retina. The second point is that the sample size between this study (n = 46 cells) and ours (n = 7760 cells in just NR condition, n = 24,152 cells when combining conditions for figure 4) are different. Similar work to map out direction selectivity (Sabbah et al., 2018; Tiriac et al, 2022) also required large sample sizes to demonstrate that the preferred directions changed as a function of retinal location. Furthermore, until the Sabbah et al. 2018 direction selectivity mapping study, patch recordings across the surface of the retina also did not report region-specific differences in direction selectivity.

From Nath & Schwartz, 2016:

2.) For the reasons given in the previous point, the asymmetries observed by Vita et al. likely, at least in part, reflect differences in the orientation preferences of non-OS RGC types. This could still be interesting, but I am concerned that the observed maps could be an artifact of the transition from the retina in vivo as a hemisphere to a flattened preparation with relieving cuts. To overcome these concerns, one or both of the following would be needed:

The concern about artifact of transitioning from hemispheric retina to flattened retina was raised with mapping direction selectivity and was addressed in both Sabbah et al, 2016 and Tiriac et al., 2022. These studies utilized independent approaches to demonstrate that relief cuts alone cannot explain the fact that preferred direction, and in our case preferred orientations, change as a function of retinal location.

In this paper, we show that even within a quadrant of the retina (thus no relief cuts between FOVs), the preferred maps of OS cells deviate as a function of retinal location (see figures 4D, 4F, and 4G).

2a.) The authors suggest that orientation preferences are aligned with concentric ellipses centered in the ventral retina. According to this model, the variation in orientation preferences in the dorsal retina should deviate from that in the ventral retina (where the ellipses are centered) in a predictable way. This prediction should be tested in recordings from the dorsal retina.

The dataset that was used to collect this data had to utilize UV stimulation due to the build of the microscope, which would only work in ventral retina, where UV cones are present. Nonetheless, we

analyzed an extra set of data from 22 FOVs and 3 mice, resulting in 1891 OS cells in temporal retina, including some dorsal areas. This serves the same purpose that the reviewer requested, to test our model in a predictable way, and indeed we found

an overrepresentation of vertical orientations (as predicted by the model). Lastly, Figures 4D-G now provides 4 separate lines of evidence that test and support our model.

2b.) Alternatively, and arguably most convincing, would be to show that these orientation maps exist in vivo. This could be tested either by two-photon imaging or electrophysiological recordings of RGC axons in retinorecipient targets. Some datasets that may allow such analyses have already been published (e.g., Liang et al. Cell 2018, Sibille et al. Nat Commun 2022).

Indeed, we believe that these orientation maps do exist in vivo, as shown by a recently published paper that addresses the reviewer's request (Malmazet et. al, 2024). In this paper, the authors image both retinal boutons and collicular neurons in the superior colliculus as they present moving bars. They find that OS preferred responses in RGC axons and SC neurons match, and both show similar concentric organization mapping. Specifically, within a localized area of the SC, OS cells prefer similar orientations and this overrepresented orientation changes as a function of SC/retinotopic/visual space. We think this aligns well with our ex vivo results in the retina. We added this reference to both the introduction and discussion of our paper.

Minor comment:

1.) In Figure 2D, please sort the neurons within each type by their orientation preferences.

We have fixed figure 1D based on this request

Reviewer #3 (Remarks to the Author):

In this manuscript, the authors investigate the orientation selectivity (OS) of retinal ganglion cells (RGCs) using an existing dataset originally generated by Tiriac et al. in 2022. They identify three functional OS RGC subtypes, with the rare OFF-OS cells exhibiting the highest selectivity among them. The study analyzes data from normally reared, dark-reared, and genetically retinal wave-affected mice, revealing robust OS development across all conditions. Additionally, the authors find differences between the global maps of OS and direction selectivity (DS).

While this manuscript presents a well-executed study on RGC orientation selectivity and provides valuable insights into RGC information processing, I have some concerns regarding its suitability for publication in a journal with a broad readership, at least with its current form. The manuscript appears somewhat incremental, with limited conceptual advancement beyond the authors' 2022 paper, the work by Baden et al. in 2016, and etc. Thus, it requires substantial revisions and expansion to enhance its significance and clarity for a broader readership. Here are my recommendations:

1. Conceptual Advancement and Discussion:

- The manuscript, apparently initially prepared as a brief report, lacks substantial expansion and discussion. The authors should delve into the biological advantages of having an overrepresented OS axis slightly offset from the DS axis. Furthermore, the implications of OFF-OS cells being more selective than ON-OS cells should be thoroughly discussed.

- The authors should consider the broader context of cortical OS/DS development, including previous work on experience-independent OS development in the visual cortex.
- While stating "implications for understanding how OS is organised downstream of the retina" (line 24), the authors omitted several crucial prior studies on experience-independent OS development in the visual cortex. For instance, Ko, Mrsic-Flogel et al. (2013), Hagihara, Ohki et al. (2015), and Hillier, Roska et al. (2017) have made significant contributions in this regard. Particularly, Hillier et al. extensively discussed the substantial influence of retinal DS on cortical DS/OS. It is essential for the authors to thoroughly examine their findings and their broader implications within the context of cortical OS/DS, encompassing not only colliculus-related research but also these noteworthy cortical studies.
- The retina has proven to be an excellent model for molecular and transcriptomic studies focused on specific cell types, and among these, DS cells have exemplified the convergence of molecular identity and function remarkably (Kim, Sanes et al. 2008; Yonehara, Roska et al. 2009, etc). It would be highly beneficial for future research to include a discussion on OS cells, particularly OFF cells, as they present an avenue for further exploration in this context.

Based on the reviewer's feedback, we have extensively added to both the introduction and discussion of the manuscript that covers all of the above points. Specifically, we have added information about the possible behavioral relevance of the OS map. Although we can only speculate at this point, we added a discussion of current understandings of the behavioral relevance of the DS maps. We also added more text in the introduction and discussion about OS development in both the SC and cortex, as well as the organization of OS in the adult SC and cortex. We now discuss prior studies on the experience-dependence and independence of OS development in visual cortex. We omitted discussing the Hillier, Roska, et al. paper since only cortical DS is assessed there (not OS). We also discuss the importance of future studies combining molecular identity with functional assessment.

2. Methodological Clarity:

The methods section could benefit from greater clarity. Specific details, such as the time-windows used for quantifying visual responses and the choice between peak dF/F and area-under-curve for responses, should be clearly outlined. Robustness against parameter selections should be demonstrated.

We chose peak dF/F because it was the parameter that was used for previous studies using data collected from a similar setup (Bos et al., current biology, 2016 & Tiriach et al., cell reports, 2022). That said, we ran an analysis using the area under the curve parameter. Although the area under the curve seems to be more conservative with identifying OS cells, the overall the results using either parameters lead to the same conclusions (3 subtypes, proportion and tuning match, and mapping matches perfectly). We leave the analysis here as a reviewer figure since the paper is already dense as it is. If requested, we can add it as a supplementary figure.

Reviewer figure 1: Peak and area calculations lead to the same general conclusions. **A.** Proportion of OS cells per FOV in ventronasal retina using either the peak or area under the curve parameters. **B.** Functional subtypes identified using either approach. **C.** Proportion and tuning characteristics of the different functional subtype using either approach. **D.** Mapping of all OS cell (top) and all the functional subtypes using either approach.

3. Inclusion of OS-DS Population:

Throughout the study, the authors treat OS cells and DS cells as mutually exclusive populations, a standard approach in the field for functional cell type classification. However, in terms of information representation and processing, disregarding the orientation selectivity of DS cells can be misleading. Indeed, some DS cells exhibit highly-tuned orientation selectivity (Figure 1C). The authors should investigate whether including this OS-and-DS population in the analysis of OS cells leads to different outcomes.

Our analysis did include the OS-and-DS population (we accepted any cells that were statistically-significantly OS without regards for DS, it just so happens that they form distinct clusters). What we tried to say in Figure 1C is that there are two clear clusters, one belonging to OS cells and another to DS cells. That said, it is true that subset of OS cells are also statistically significantly DS (3%) and a subset of DS cells are statistically significantly OS (8%). When looking at the whole population (including non-OS or DS cells), these OS and DS cells account for <1% of the total cell population.

The new Figure S5 (discussed more in point below) performs an analysis that compares mapping between OS but not DS cells and all OS cells. As expected, since only 3% of OS cells are statistically significantly OS, the distributions are similar.

Also, the manuscript lacks clarity regarding data inclusion criteria. If OS and DS cells are separately used to produce maps in analyses for Figure 3, the authors should explore the effect of including the OS-and-DS population cells or all visually responsive cells on OS/DS map axes analysis.

Yes, as described above, cells that were both OS and DS were not excluded from the analysis but were rare (3% of OS cells). We also performed an analysis on all visually responsive neurons and all visually responsive neurons – OS cells. We now include a new Figure S5 that contains all of this information.

4. Neuropil Contamination:

Considering that calcium imaging was conducted using chemical AM dyes, it is expected that the dataset may include considerable neuropil contamination owing to tissue scattering. Such contamination can potentially diminish orientation selectivity (OS) by adding non-selective signals to actual selective signals. Given the sparse presence of OFF responsive cells, it is plausible that OFF cells are less susceptible to this signal contamination when compared to ON OS cells. To validate that the tuned OS observed in OFF cells is not simply an artefact, the authors should systematically vary the degree of neuropil signal subtraction, as demonstrated in studies such as Kerlin, Reid et al. (2010) and Hagiwara, Ohki et al. (2015).

The papers cited here are in cortex, where there is a lot of neuropil space between cells. In fact, in the methods section of these papers to perform these analyses, it is stated to use ROIs that are just outside of cell somas. In the ganglion cell layer, there is no space between cells, and thus this analysis, as far as we know, is not possible in the retina.

Other minor points:

We fixed all of these issues and we hope that the new Figure 2 and revised Figure 4 help with visually illustrating the extent of the overrepresentation.

1. The title of Figure 1 is somewhat indirect. Consider a more direct description to convey the greater tuning of OFF OS cells.
2. In line 102, "axed"
3. In Figure 1, please clarify the correspondence between B and A.
4. Consider including pixel-based orientation/direction color-map examples in Figure 3 to visually illustrate the extent of overrepresentation.
5. In Figure 1, for panels G and H, "direction" should be replaced with "orientation."

REVIEWER COMMENTS

Reviewer #1 (Remarks to the Author):

The revised paper looks vastly improved and I believe this study will contribute significantly to our growing knowledge of OS mechanisms and retinal function. The authors have responded satisfactorily to my comments. However I do have one further question/suggestion. I appreciate the new set of experiments performed in *vglut2-cre:Gcamp6s* mouse line and the data looks convincing. I believe the authors have imaged both ventronasal and ventrotemporal retinal regions in this mouse line. If yes, the authors should have a supplementary figure showing the spatial organization of OS map in this line is similar to that observed in Figure 2.

Reviewer #1 (Remarks on code availability):

I browsed through the code and it looks fine. I did not have time to install and run the code.

Reviewer #2 (Remarks to the Author):

The authors have satisfactorily addressed my previous concerns. Congratulations on an exciting study.

Reviewer #3 (Remarks to the Author):

While I acknowledge the improvements made in the manuscript, I must note that the authors have yet to address two previous concerns regarding the validity of the signals. Consequently, I maintain my stance that the main conclusions lack full support from the data.

1: Neuropil contamination

The contamination source may not exclusively be neuropil but could also involve surrounding cells in scattering tissues like the retina. This is particularly notable as the authors mention, "In the ganglion cell layer, there is no space between cells," alongside low-resolution imaging. Notably, the segmented cells in Fig. 1B appear to highly overlap, suggesting potential cross-contamination between neighboring cells. Therefore, the overrepresentation of orientation, as claimed by the authors, might simply result from cross-contamination. I'm not insisting on adopting the exact decontamination approach seen in cortical imaging papers. However, the authors should ensure the robustness of their conclusion through additional analyses.

2: Pixel-based orientation selectivity map

In response to my previous comments, the authors mentioned: "We fixed all of these issues and we hope that the new Figure 2 and revised Figure 4 help with visually illustrating the extent of the overrepresentation." However, I couldn't find any pixel-based map(s) in the revised manuscript, which is disappointing. Unlike heavily processed analyses, a pixel-based map offers an excellent visualization of the robustness of selectivity in individual cells. This is especially crucial considering the aforementioned cross-contamination issue. If there was any ambiguity regarding the term "pixel-based map," please refer to Ohki, Reid et al., 2005, Fig1c vs. Fig1e.

REVIEWER COMMENTS

Reviewer #1 (Remarks to the Author):

The revised paper looks vastly improved and I believe this study will contribute significantly to our growing knowledge of OS mechanisms and retinal function. The authors have responded satisfactorily to my comments. However I do have one further question/suggestion. I appreciate the new set of experiments performed in *vglut2-cre:Gcamp6s* mouse line and the data looks convincing. I believe the authors have imaged both ventronasal and ventrotemporal retinal regions in this mouse line. If yes, the authors should have a supplementary figure showing the spatial organization of OS map in this line is similar to that observed in Figure 2.

We thank the reviewer for the kind remarks.

The GCaMP dataset was mostly collected in directly ventral and temporal quadrants of the retina. We did have one mouse where the retinal preparation allowed for imaging in ventrotemporal retina and analyzed it to address this comment. Data come from 1 mouse, 4 fields of views, and represents 293 OS cells. We included this as part of supplementary figure 7, which contained other supplementary information about the *vglut2cre::GCaMP6s* dataset.

Reviewer #1 (Remarks on code availability):

I browsed through the code and it looks fine. I did not have time to install and run the code.

Reviewer #2 (Remarks to the Author):

The authors have satisfactorily addressed my previous concerns. Congratulations on an exciting study.
Thank you!

Reviewer #3 (Remarks to the Author):

While I acknowledge the improvements made in the manuscript, I must note that the authors have yet to address two previous concerns regarding the validity of the signals. Consequently, I maintain my stance that the main conclusions lack full support from the data.

Our apologies for missing these two points. We now have a new Figure S2 that addresses both points and we provide step-by-step explanations below.

1: Neuropil contamination

The contamination source may not exclusively be neuropil but could also involve surrounding cells in scattering tissues like the retina. This is particularly notable as the authors mention, "In the ganglion cell layer, there is no space between cells," alongside low-resolution imaging. Notably, the segmented cells in Fig. 1B appear to highly overlap, suggesting potential cross-contamination between neighboring cells. Therefore, the overrepresentation of orientation, as claimed by the authors, might simply result from cross-contamination. I'm not insisting on adopting the exact de-contamination approach seen in cortical imaging papers. However, the authors should ensure the robustness of their conclusion through additional analyses.

We have performed a new analysis to test for cross-contamination between nearby ROIs. For all OS cells, we identified the 6 nearest ROIs to that cell (we chose 6 because a hexagonal matrix provides the best packing factor and matches with the packing of cells in the ganglion cell layer; Figure S2D). We also performed the same analysis on a similar sample of cells that were randomly picked throughout the dataset (denoted as R). If cross contamination exists, there should be a large number of cells around OS cells that are also OS. Although we saw a slightly larger amount for the OS dataset (mean±stdev = 30.62±3.0%), this number did not deviate greatly from the random dataset (mean±stdev = 25.44±2.7%) and thus cannot account for the overrepresentation that we reported (Figure S2E).

We modeled how cross-contamination would affect the analysis performed in S2D and E (Figure S2F). We used a sample mask of ROIs from one of our dataset ("All ROIs") and simulated three distributions of cells with varying degrees of cross-contamination. First (1), we randomly sampled 25% of ROIs to be OS, resulting in a distribution that does not force cross-contamination. Second (2), we forced varying levels of cross-contamination (1-7 neighbors per cluster of ROIs) while still making sure that OS cells accounted for 25% of ROIs. Third (3), we repeated step two but now forced every cluster to include 7 ROIs. We repeated the calculations in D-E for 1000 simulations to obtain the distributions found at right in F (mean±stdev: 1: 24.87±1.8%; 2: 60.40±3.1%; 3: 70.60±2.7%). The random sampling of OS cells (1) resulted in values similar to our dataset (S2E) whereas forcing cross-contamination greatly increased the likelihood that neighboring ROIs are OS.

Lastly, we asked whether nearby OS cells in both our analysis anchored on OS and R cells would exhibit different distributions of preferred orientations, the idea being that if cross contamination occurred, this should skew the histograms in the favor of the overrepresented orientations. Analysis in both ventronasal and ventrotemporal retina exhibit similar distributions of preferred orientations in nearby OS cells for both the data anchored on OS and R cells (Figure S2G).

2: Pixel-based orientation selectivity map

In response to my previous comments, the authors mentioned: "We fixed all of these issues and we hope that the new Figure 2 and revised Figure 4 help with visually illustrating the extent of the overrepresentation." However, I couldn't find any pixel-based map(s) in the revised manuscript, which is disappointing. Unlike heavily processed analyses, a pixel-based map offers an excellent visualization of the robustness of selectivity in individual cells. This is especially crucial considering the aforementioned cross-contamination issue. If there was any ambiguity regarding the term "pixel-based map," please refer to Ohki, Reid et al., 2005, Fig1c vs. Fig1e.

Again, we apologize for missing this comment. We struggled with this one since the selectivity of the representative cells in cortex are so tuned (see Fig 1D for the Ohki et al paper) compared to the tuning of OS cells in our dataset. Nonetheless, we built the requested analysis and generated a pixel-based map where the color represents the preferred orientation and the hue represents the strength of the tuning. Without any processing, the resulting image is quite noisy (Fig S2A). Applying a spatial median filter to remove solitary pixels (i.e., pixels without any neighboring pixels of similar preferred orientation) reveals the clusters of similar-orientation-preference pixels that are putative OS cell (Fig S2B). Statistically-significant OS cells from our analysis map well on this pixel-based map (Fig S2C) though some discrepancies remain, likely due to the fact that our analysis uses both a quality index threshold (cells

must have consistent responses to repeated trials) and a permutation test to determine statistically-significant OS cells. Importantly, the pixel-based analysis demonstrates the same overrepresentation of one orientation that our analysis also reports.

REVIEWERS' COMMENTS

Reviewer #3 (Remarks to the Author):

I am satisfied with the additional analyses provided by the authors. They would be informative for readers to interpret/evaluate the data presented in the paper.